# Genetic Diversity Assessment and Core Germplasm Screening of Blackcurrant (*Ribes nigrum*) in China via Expressed Sequence Tag–Simple Sequence Repeat Markers

**DOI:** 10.3390/ijms26052346

**Published:** 2025-03-06

**Authors:** Xinyu Sun, Qiang Fu, Dong Qin, Jinyu Xiong, Xin Quan, Hao Guo, Jiahan Tang, Junwei Huo, Chenqiao Zhu

**Affiliations:** 1College of Horticulture & Landscape Architecture, Northeast Agricultural University, Harbin 150038, China; sunxinyuly@163.com (X.S.); qw602002141@163.com (Q.F.); dongq9876@126.com (D.Q.); 13277283006@163.com (J.X.); 15846486578@163.com (X.Q.); gh15184224644@outlook.com (H.G.); 2Mudanjiang Branch Institute, Heilongjiang Academy of Agricultural Sciences, Mudanjiang 157041, China; 18145310218@163.com

**Keywords:** *Ribes nigrum*, EST-SSR markers, genetic diversity, population structure, core germplasm resources

## Abstract

Blackcurrant (*Ribes nigrum* L.) has high nutritional value for human health due to its abundant vitamin C, flavonoids, and organic acids. However, its breeding and genetic research have been severely hindered by the lack of scientific tools such as molecular markers. Here, we identified 14,258 EST-SSR loci from 9531 CDS sequences with lengths greater than 1 kb, which comprised 6211 mononucleotide repeats, 4277 dinucleotide repeats, and 2469 trinucleotide repeats. We then randomly selected 228 EST-SSR loci for PCR amplification and gel electrophoresis imaging in the *Ribes* collection of Northeast Agricultural University (95 blackcurrant cultivars and 12 other *Ribes* accessions). As a result, 31 pairs of markers produced clear and reproducible bands of the expected size. Based on the 107 *Ribes* accessions, the allele number (*Na*), information index (*I*), observed heterozygosity (*Ho*), expected heterozygosity (*He*), and polymorphic information content (*PIC*) of the 31 markers were 2–5, 0.23–1.32, 0.07–0.71, 0.11–0.68, and 0.14–0.67, respectively. For the blackcurrant gene pool, neighbor-joining and population structure analysis revealed three clusters, which did not align well with their geographical origins. Based on the results, two sets with 21 and 19 blackcurrant cultivars were identified by Power Core (PC) and Core Hunter (CH) programs. The integrated core germplasm (IC) set with 27 cultivars derived from the PC and CH sets harbored abundant genetic diversity, where the allele retention rate accounted for 98.9% of the blackcurrant gene pool. The SSR markers, data, and core germplasms presented in this study lay a solid foundation for the phylogenetic study, molecular breeding, and conservation genetics of *Ribes*, especially *Ribes nigrum*.

## 1. Introduction

Blackcurrant (*Ribes nigrum* L.) is the most economically significant species within the *Ribes* genus of the Grossulariaceae family [1]. The annual global production of blackcurrant-dominated currants reaches 764,499 tons [2]. Blackcurrant is rich in bioactive compounds including anthocyanins, flavonoids, polysaccharides, vitamins, and organic acids [3,4,5,6,7]. Additionally, its seed oil contains unsaturated fatty acids such as alpha-linolenic acid and gamma-linolenic acid, as well as phytosterols [8,9]. These functional attributes offer distinctive pharmacological properties in reducing the incidence of type 2 diabetes, lowering blood lipids, protecting vision, exhibiting antiviral and anticancer effects, and enhancing immune function [10]. Data have shown that currants, particularly blackcurrants, are cultivated in 37 countries, among which the Russian Federation and Poland account for 49.89% and 32.24% of the total cultivation area, respectively [2,11]. The cultivation of blackcurrants in China can be traced back to as early as 1917 [12,13]. Elite blackcurrant cultivars were initially introduced to Northeast China by Russian immigrants from the northern side of the Amur River, resulting in a cultivation history spanning over one century. With the rapid development of China’s light industry after 1958, blackcurrant fructose and fruit wine have gained increasing popularity in both domestic and international markets, which has significantly increased the demand for blackcurrant raw materials, leading to a steady expansion of the cultivation area. By 1985, the blackcurrant cultivation area in Heilongjiang and Jilin provinces reached 21,940 hectares [14], which triggered great efforts in the selection and breeding of blackcurrant cultivars with desirable traits. In 1986, some Chinese breeding institutions, such as Northeast Agricultural University (NEAU), introduced elite blackcurrant germplasms from foreign countries including Russia, Poland, and Sweden. During the past 70 years, a total of 95 blackcurrant cultivars and other 12 *Ribes* accessions have been collected and preserved in Northeast Agricultural University (hereinafter referred to as NEAU collection). By crossbreeding these foreign germplasms with the germplasms native to China, breeders have successfully developed high-yielding, cold-resistant, and superior-quality cultivars such as ‘Daisha’, ‘HanFeng’, and ‘Danjianghei’ [14,15,16]. However, the genetic diversity level and population structure of Chinese blackcurrant germplasms remain unknown due to the limited original information, incomplete or missing breeding records, possible repeated introduction of identical germplasms, and the involvement of local wild species, which has severely limited the further breeding and improvement in blackcurrant.

Genetic diversity assessment of plant germplasm can be conducted based on morphological traits, anatomical and biochemical markers, and molecular markers [17]. Molecular markers offer greater advantages over other methods due to their independence from complex environmental factors and phenological stages, high repeatability, and simplicity and speed of sampling [18]. Since 1995, molecular markers such as RAPD (random amplified polymorphic DNA), AFLP (amplified fragment length polymorphism), and ISSR (inter-simple sequence repeat) have been extensively utilized to assess the genetic diversity of different blackcurrant germplasms [19,20,21,22]. However, the further development of molecular markers was impeded by high cost, poor reproducibility, and low polymorphism. In 2002, Brennan and Jorgensen developed SSR (simple sequence repeat) markers by using the microsatellite library enrichment method for blackcurrants [23]. These markers offer high diversity, excellent reproducibility, and ease of use, and have been employed in the analysis of genetic diversity in blackcurrant populations and the identification of related species. However, the cost of developing related SSR markers was not reduced until the development of NGS (next-generation sequencing) technology [24,25]. To date, the development of SSR markers using genomics and EST (expressed sequence tag) has been applied to genetic research on berries such as *Vaccinium* [26], *Rubus* [27], and *Vitis* [28]. In addition, on the basis of an existing collection of blackcurrant resources and breeding, it is equally important to establish a core germplasm set for the better utilization of blackcurrant resources and the reduction of cost of research and breeding, which can be accomplished by the application of SSR markers [29,30].

This study utilized blackcurrant transcriptome data to develop EST-SSR markers, aiming to (1) develop new SSR markers that are universally applicable to crops within the *Ribes* genus, (2) analyze the genetic diversity levels of existing blackcurrant cultivars in China, (3) establish core germplasm set as a material basis for breeding and selection efforts, and (4) assess the phylogenetic relationships between blackcurrant and other *Ribes* accessions, providing a database for the intraspecific genetic classification of *Ribes*.

## 2. Results

### 2.1. Identification of SSR Loci

SSR loci were identified for transcriptomic data on blackcurrant fruit development (PRJNA1021373) [31]. Among the 23,220 assembled unigenes with lengths exceeding 1 kb, 9531 (41.0%) of them were found to harbor a total of 14,258 SSR loci, which were predominantly composed of mononucleotide repeats (6211), dinucleotide repeats (4277), and trinucleotide repeats (2469) (Figure 1A, Appendix A). In total, 98 motif types were identified among the EST-SSR loci in addition to the presence of compound repeats. The largest number of mononucleotide repeats was A/T (6062), followed by three dinucleotide repeats, including AG/GA (1621), CT/TC (1600), and AT/TA (683). The three most frequent trinucleotide repeats were CTT/GAA (286), AGA/TCT (249), and AAG/TTC (242) (Figure 1B). Except for the compound-type EST-SSR loci, other loci showed lengths ranging from 10 to 72 bp, with 12 bp loci (793 mononucleotide repeats and 1497 dinucleotide repeats) being the most prevalent loci, followed by 15 bp loci (387 mononucleotide repeats and 1588 trinucleotide repeats), and 14 bp loci (447 mononucleotide repeats and 1184 dinucleotide repeats) (Appendix A).

### 2.2. EST-SSR Marker Development and Their Effectiveness Among Different Cultivars

A total of 228 EST-SSR loci were randomly selected, and primers were designed and synthesized to test their amplification on the 107 *Ribes* accessions. Eventually, 31 pairs of primers (13.6%) could amplify unambiguous bands (Table 1). The number of effective alleles (*Ne*) varied from 1.13 to 3.12, with a mean value of 1.83; the Shannon information index (*I*) varied between 0.23 and 1.32, with a mean value of 0.71; the observed heterozygosity (*Ho*) ranged from 0.07 to 0.71, with an average value of 0.33; the expected heterozygosity (*He*) values varied from 0.11 to 0.68, with a mean value of 0.40. Polymorphic information content (*PIC*) varied from 0.14 to 0.67, with a mean value of 0.40. Ten primers exhibited high polymorphisms (*PIC* > 0.5), among which S-18 exhibited the highest polymorphism. There were 14 primers showing moderate polymorphisms (0.25 ≤ *PIC* ≤ 0.5). Seven primers showed low polymorphisms (*PIC* < 0.25), in which S-172 exhibited the lowest polymorphism.

### 2.3. Genetic Assessment via Neighbor-Joining Cluster Analysis

Genetic relatedness among the 95 blackcurrant cultivars was evaluated using 31 pairs of EST-SSR markers and analyzed with NJ (neighbor-joining) cluster analysis (Figure 2). The NJ cluster analysis classified the 95 blackcurrant cultivars into three clusters, which was not totally consistent with their origins (Figure 2A). Cluster 1 comprised two sub-clusters, with Sub-cluster 1 (purple branch) including nine exotic cultivars and eight Chinese cultivars, while Sub-cluster 2 (pink branch) consisting of 28 cultivars, all of which were from China. Cluster 2 contained 13 exotic cultivars and 11 Chinese cultivars. Cluster 3 included 26 cultivars, half of which were introduced from foreign countries. ‘Baopifengchan’ in Sub-cluster 1 was the most genetically distant from the other 94 cultivars, with a genetic distance of 0.29.

To elucidate the genetic relationships among the 107 *Ribes* accessions, an NJ tree based on 31 EST-SSR markers was constructed to calculate the genetic distances. The results showed that these *Ribes* accessions could be divided into two main branches, with one branch including seven cultivars, all of which belong to *R. rubrum*, while the other branch includes 100 cultivars, which belong to the remaining six species (Figure 2B). Among the 100 cultivars in the second branch, two cultivars, ‘Xinganchabiao’ (*R. panciflorum*) and ‘Ussuri’ (*R. ussuriensis*), were grouped together with 95 cultivars of *R. nigrum*, probably due to gene introgression caused by the use of *R. nigrum* in crossbreeding trials. For the remaining three cultivars, ‘Hongyeheidou’ (*R. americanum*) was classified as a separate branch, while ‘Pixwell’ (*R. uva-crispa*) and ‘Xiangchabiaozi’ (*R. odoratum*) formed sister branches to each other. Within the blackcurrant gene pool, the 95 cultivars generally exhibited the same affinities as shown in Figure 3A. Nine cultivars in Sub-cluster 1 had a close distance to other *Ribes* species, with the ‘Sophia’ (*R. nigrum*) cultivar being the sister of ‘Ussuri’ (*R. ussuriensis*).

### 2.4. Genetic Assessment via Population Structure Analysis

Among the 95 blackcurrant cultivars collected and bred, there were twenty Russian cultivars, four Polish cultivars, seven British cultivars, one Swedish cultivar, one Dutch cultivar, one Danish cultivar, one Canadian cultivar, and sixty Chinese cultivars. These cultivars from eight sources were used for population structure analysis. The log-likelihood of the structure analysis indicated an optimal K-value of 3 (K = 3), suggesting that the blackcurrant gene pool could be divided into three clusters (Figure 3A). Cluster A had a total of thirty-three cultivars, including eleven Russian cultivars, three Polish cultivars, two British cultivars, one Swedish cultivar, and sixteen Chinese cultivars. Cluster B had thirty-one cultivars, including eight Russian cultivars, four British cultivars, one Polish cultivar, one Danish cultivar, one Dutch cultivar, one Canadian cultivar, and fifteen Chinese-selected cultivars. Cluster C contained thirty-one cultivars, except for ‘Vologda’ introduced from Russia and ‘Ben Lomond’ introduced from the United Kingdom, and the remaining twenty-nine cultivars were all from China (Figure 3B, Appendix A). The AMOVA among clusters at K = 3 showed that genetic variation occurred predominantly between cultivars within the blackcurrant cultivars (81%) (Table 2). The blackcurrant gene pool at K = 5 also showed a peak, though the size was smaller than that observed at K = 3 (Figure 3C, Appendix A). Cluster D contained fourteen cultivars, including one Polish cultivar, six Russian cultivars, and seven Chinese cultivars. Cluster E comprised seventeen cultivars, including five Russian cultivars, two British cultivars, one Swedish cultivar, and nine Chinese cultivars. Cluster F had fifteen cultivars, including one Polish cultivar, five Russian cultivars, and nine Chinese cultivars. Cluster G contained five British cultivars, four Russian cultivars, two Polish cultivars, one Canadian cultivar, one Danish cultivar, one Dutch cultivar, and eight Chinese cultivars. Cluster H included twenty-seven cultivars, all of which were selected and bred in China.

We further investigated the population structure of 107 *Ribes* accessions. The highest peak (K = 2) appeared at ΔK = 657.28, indicating that the tested accessions can be divided into two clusters (Appendix A). Cluster I (98 accessions) includes all accessions of *R. nigrum* (95 cultivars), *R. ussuriensis* (1 accession), *R. uva-crispa* (1 accession), and *R. panciflorum* (1 accession), while Cluster II includes all accessions of *R. rubrum* (7 accessions), *R. americanum* (1 accession), and *R. odoratum* (1 accession) (Appendix A). Interestingly, another peak (K = 7) appeared at ΔK = 85.48, where *R. rubrum* (Cluster VII, 7 accessions) forms a distinct group separate from the other 100 *Ribes* accessions (Appendix A). The results were consistent with the phylogenetic analysis (Figure 2B).

### 2.5. Establishment of a Core Germplasm Repository

To better conserve the genetic diversity of the NEAU collection and simultaneously provide instructive information for subsequent breeding programs, a core germplasm set was established using Power Core v 1.0 (PC) and Core Hunter (https://www.corehunter.org/, accessed on 26 May 2024) (CH) programs based on genetic markers, matrices, and genetic parameters of populations. The sample size and genetic diversity parameters for each set are shown in Table 3. The PC set contained 21 (22.1%) blackcurrant cultivars, while the CH set contained 19 (20.0%) cultivars. A comparison of the genetic diversity and retention rate of the two sets revealed that the PC set had higher values of *I*, *He*, and *PIC* and lower values of *Ne* and *Ho* than the CH set. The two sets included thirteen repetitive cultivars, while there were eight cultivars unique to the PC set and six cultivars unique to the CH set (Figure 4A, Appendix A). To ensure the representativeness of the core germplasm resources, the two sets obtained with different methods were combined to create an integrated core germplasm (IC) set. The IC set included one Swedish cultivar, four Russian cultivars, four Polish cultivars, and eighteen Chinese cultivars. A comparative analysis of genetic diversity parameters revealed that the IC set exhibited significantly higher genetic diversity than the PC and CH sets. The principal coordinate analysis (PCoA) results indicated that the IC set could better represent the initial collection than the PC and CH sets, which is consistent with the initial set in terms of genetic distribution (Figure 4B–D).

## 3. Discussion

### 3.1. Genetic Diversity and Genetic Structure in Blackcurrant

The earliest study by R. Brennan utilized the microsatellite-enriched libraries method to develop 11 simple sequence repeat (SSR) markers, achieving diversity levels ranging from 0.18 to 0.91. Due to the high cost and experimental difficulty of this method, it was not subsequently adopted for further marker development [23,32]. The study of SSR markers in blackcurrants did not gain popularity until Joanne R. Russell and her colleagues from the James Hutton Institute in the UK applied transcriptome-based second-generation sequencing (2GS) technology to develop 3000 pairs of expressed sequence tag–simple sequence repeat (EST-SSR) markers [33]. However, there are fewer new SSR markers for genetic identification, mostly following the already developed markers (e.g., RJL1-11, e1-O01, g2-G12, etc.) for analysis [34,35,36]. This study utilized newly developed EST-SSR markers to investigate blackcurrant cultivars collected and cultivated in China for over a century, and for the first time explored primer polymorphism. The mean values of *He* and *Ho* for the newly developed EST-SSR markers are lower than those for the SSR markers [23,37]. On the one hand, frequent gene exchange among experimental materials is observed during the process of breeding new varieties. On the other hand, the relatively low variation in EST markers developed during genetic evolution may also be a contributing factor. However, 10 pairs of the developed primers still demonstrate high representativeness (PIC > 0.5). The newly developed markers provide an important tool for further studying the genetic diversity of blackcurrant cultivars, laying a foundation for future cultivar improvement and genetic research.

Genetic polymorphism is ubiquitous and necessary in the process of species evolution, which is formed under the joint action of biological factors such as gene mutation and gene migration and abiotic factors such as climate change and geographical isolation [38,39]. The earlier genetic diversity of blackcurrant was determined based on phenological periods, plant morphological identification, yield, fruit quality, and resistance studies [40]. However, the application of this method is restricted by the influence of the environment on phenotypic traits and the limited morphological variation. In contrast, genetic diversity analysis based on markers such as SSR and SNP allows the more accurate identification and analysis of the diversity of different blackcurrant cultivars. Among them, the ‘Baopifengchan’ cultivar is extremely resistant to cold and powdery mildew, presumably domesticated from a wild variety in Russia. In its original sampling place, other Russian cultivars have natural geographic isolation, less gene exchange, and more genetic distance. The results of population structure analyses with different methods were inconsistent with the geographical origins of blackcurrant. Over the past century, introgression of blackcurrant gene pool from different origins and genetic improvement by breeders have increased gene flow, thereby reducing the geographic differentiation in their distribution. In addition, we corrected the information on three cultivars that were repeatedly introduced at different times, ‘Ojebyn’, ‘Fertodi’, and ‘Hanfeng’, through the results of the trial.

### 3.2. Construction of Core Germplasm Resources

The construction of core germplasm sets is essential for crops with high economic value and wide cultivation areas to maximize the conservation of genetic diversity while minimizing genetic redundancy [41,42,43]. Researchers can use phenotypic data, geo-environmental data, agronomic trait data, genotypic data, genetic diversity data, and other data for the construction of core germplasm resources [44,45]. For instance, the RIBESCO project, launched from 2007 to 2011, aimed to establish a core germplasm collection of *Ribes* through phenotypic and molecular characterization, thereby enhancing the informational and safety standards of *Ribes* genetic resource repositories [46]. This study has the same purpose as the RIBESCO project: to strengthen the characterization and conservation of the *Ribes* genus, particularly the blackcurrant germplasm, as well as to promote the transfer and use of genetic materials from the *Ribes* genus. In this study, the genotyping data obtained by EST-SSR markers were used to initially construct core germplasm resources, and a core germplasm resource set containing 27 (28.7%) blackcurrant germplasms was finally constructed. The next step will involve the identification of phenotypic and agronomic traits to address the limitations of the IC set and establish a comprehensive blackcurrant core collection, which will provide more valuable information for guiding parent selection.

### 3.3. Genetic Polymorphisms Within the Ribes Genus

As economically valuable plants, some *Ribes* species possess ornamental properties (*R. alpinum*, *R. americanum*, and *R. odoratum*), medicinal value (*R. nigrum*, *R. rubrum*, and *R. uva-crispa*), and stress resistance (*R. rubrum*) [47,48,49,50]. In recent years, some research has revealed that certain species in the *Ribes* genus can be crossed with each other, not only improving the environmental resilience of the plant but also increasing the economic value of certain species [37]. The results of phylogenetic and genetic structure studies of seven different *Ribes* species using EST-SSR markers were similar to the ML evolutionary trees constructed in the previous phase by applying the whole chloroplast genome [51] and by applying chloroplast DNA simple sequence repeats (cpSSRs) [52]. In addition, the phylogenetic tree showed that cultivars of ‘Xinganchabiao’ (*R. panciflorum*) and *R. nigrum* were clustered together. Due to the small amount of materials in this experiment for *Ribes* compared with that of blackcurrant, our results may be relatively biased, and more materials are needed to further explore intrageneric differentiation. Their relationship will be further explored in the future with the joint analysis of their morphological traits.

## 4. Materials and Methods

### 4.1. Plant Materials

Over the years, 107 *Ribes* accessions were collected and selected, which are designated as NEAU collection hereafter. This collection includes ninety-five cultivars of *R. nigrum*, six accessions of *Ribes rubrum* (red currant), one cultivar of *Ribes rubrum* (white currant), one cultivar of *Ribes uva-crispa*, one cultivar of *Ribes ussuriensis*, one cultivar of *Ribes panciflorum*, one cultivar of *Ribes americanum*, and one cultivar of *Ribes odoratum*. These accessions included some imported from Russia (27 accessions), Poland (5 accessions), Britain (7 accessions), America (1 accession), Canada (1 cultivar), Denmark (1 cultivar), French (1 accession), Netherland (1 cultivar), Sweden (1 cultivar), and 62 accessions selected and bred in China over the past years (Table 4). The materials are currently grown at the Horticultural Station of Northeast Agricultural University (126.73° E, 45.74° N). Between April and June 2023, tender leaves from all materials were collected and preserved in an ultra-low temperature freezer at −80 °C.

### 4.2. DNA Extraction and Quantification

DNA extraction was carried out using the Hi-Fast Plant Genomic DNA Kit (GeneBetter BioTech Co., Ltd., Beijing, China) following the manufacturer’s instructions (http://www.gene-better.cn/) (accessed on 12 April 2023). The purity and concentration of the DNA were measured using an ultra-micro spectrophotometer (Implen N60, Munich, Germany), and the DNA samples that passed the test were diluted to 10 ng/μL and stored in a refrigerator at −20 °C.

### 4.3. Genotyping with EST-SSR Markers

The development of EST-SSR loci and markers was based on the transcriptome data of blackcurrant obtained from the NCBI database (PRJNA1021373) [31]. After assembly, unigenes with lengths greater than 1 kb were selected for subsequent analysis. MISA (MIcroSAtellite identification tool) (v1.0) [53] was used to identify EST-SSR sites with default parameters. Primer Premier 5 [54] was employed to design upstream and downstream primers for the EST-SSRs. The parameters for primer design were as follows: primer pairs with annealing temperatures ranging from 57 to 63 °C, primer lengths between 18 and 27 bp, product lengths ranging from 100 to 280 bp, and GC content between 40% and 60%.

The PCR reactions were conducted in a 20 μL reaction volume, each containing 2 μL of 10 ng/μL DNA template, 1 μL of both forward and reverse primers, and 16 μL of T3 Super PCR Mix (Beijing Tsingke Biotech Co., Ltd., Beijing, China). The PCR program was as follows: an initial denaturation at 98 °C for 2 min, followed by 26–30 cycles of denaturation at 98 °C for 10 s, annealing at 55–59 °C for 15 s, and extension at 72 °C for 15 s; the final extension was performed at 72 °C for 5 min. An electrophoresis apparatus model JY-ECP 3000 (Bio-Rad, Hercules, CA, USA) was used to perform 8% polyacrylamide gel electrophoresis for analyzing PCR products, which was silver-stained for the visualization of the bands following the previously reported protocol [55].

### 4.4. Statistical Analysis

The assessment of genetic diversity involved the use of GenAlEx (v6.502) [56] and PowerMarker (v3.25) [57] for the calculation of amplification band observed alleles (*Na*), effective alleles (*Ne*), expected heterozygosity (*He*), Shannon’s information index (*I*), and polymorphic information content (*PIC*). The population structure analysis of the accessions was conducted using Structure (v 2.3.4) [58]. The analysis of molecular variance (AMOVA) was carried out using GenAlEx (v6.502). Clustering analysis was performed using MEGA7 [59], and the resulting dendrogram was constructed using the Tree plot module. Subsequently, ITOL [60] was used to enhance the esthetics of the phylogenetic tree.

### 4.5. Construction of a Core Germplasm Repository

Core germplasm libraries were constructed using Power Core (v 1.0) [61] for M-strategy and Core Hunter (https://www.corehunter.org/, accessed on 13 June 2024) for the rapid construction of core subsets based on genetic markers, matrices, and genetic parameters of populations, evaluated using GenAlEx (v6.502) and Powermarker (v3.25). The online website https://www.chiplot.online/ (accessed on 6 July 2024) was used for PCoA (principal coordinate analysis) mapping based on Nei’s distances [62].

## 5. Conclusions

In this study, the development of universal EST-SSR markers, analysis of genetic polymorphisms, and construction of core germplasm resources were conducted for 95 blackcurrant cultivars widely cultivated in China. A total of 31 pairs of EST-SSR markers were successfully developed and applied to the genetic analysis of blackcurrant cultivars. Additionally, the 31 EST-SSR markers were successfully employed to investigate the genetic relationships within the *Ribes* genus. The results of NJ and structural analyses indicated that there was no specific correlation between different germplasms based on geographic origins. The 27 blackcurrant cultivars in the integrated core germplasm (IC) exhibited higher genetic similarity to the initial collection compared with the PC and CH sets. The results provide a tool for the further exploration of different species within the genus *Ribes*.

## Figures and Tables

**Figure 1 ijms-26-02346-f001:**
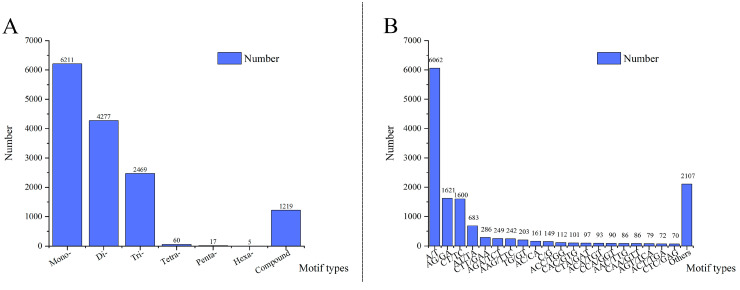
The statistical chart for the distribution of EST-SSR loci. (**A**) The distribution of EST-SSR loci with different motif types; (**B**) the distribution of EST-SSR loci in different repetitive sequences.

**Figure 2 ijms-26-02346-f002:**
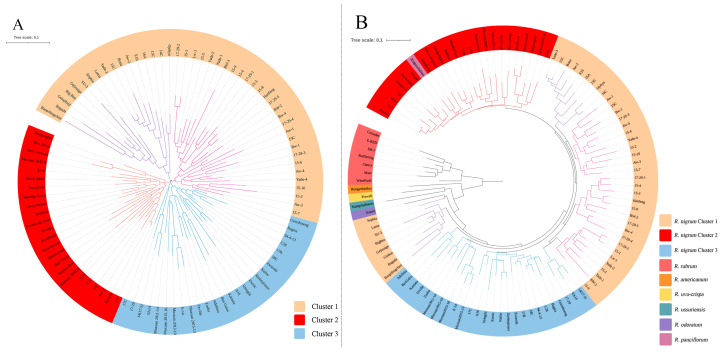
The genetic relationships of blackcurrant cultivars (**A**) and *Ribes* accessions (**B**) determined with 31 EST-SSR markers using the neighbor-joining approach.

**Figure 3 ijms-26-02346-f003:**
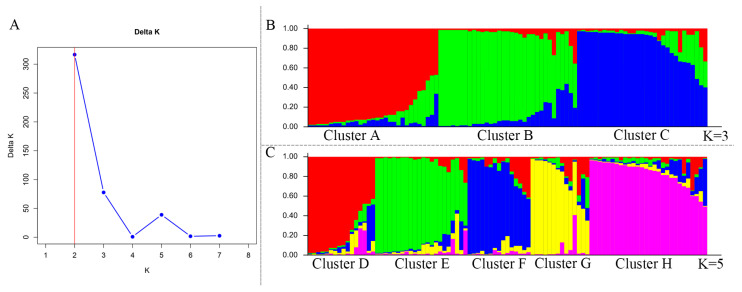
Population structure analysis of 95 blackcurrant cultivars based on 31 EST-SSR markers. (**A**) Delta K calculation was performed for each K value according to Evanno method; (**B**) population structure analysis (K = 3); (**C**) population structure analysis (K = 5).

**Figure 4 ijms-26-02346-f004:**
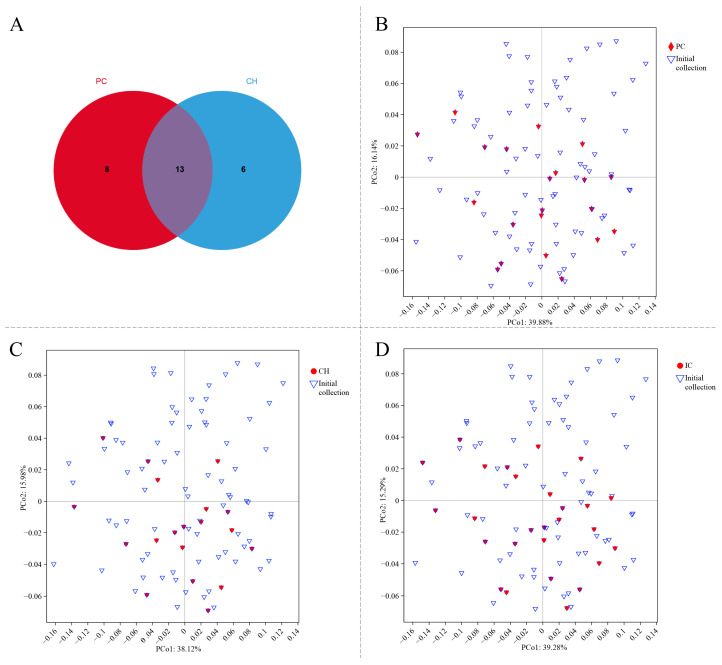
The number of cultivars in the two core germplasm sets generated by Power Core (PC) and Core Hunter (CH), and the principal coordinate analysis (PCoA) of blackcurrant gene pool based on Nei’s gene diversity index. The IC set was created by combining the PC set and CH set. (**A**) Venn diagrams of germplasm resources within the PC and CH sets; (**B**–**D**) PCoA comparison of the PC, CH, and IC sets with the initial collection.

**Table 1 ijms-26-02346-t001:** Repeat motif, primer sequence, Tm, and polymorphism information for 31 EST-SSR markers.

Primer	Report	Primer 5′-3′	Tm/°C	*Na*	*Ne*	*I*	*Ho*	*He*	*PIC*
S-3	(CCG)8	GCGAAGAAGAAGTTGATCCG GGAGGGTTCTTCGATTCACA	56.00	3	1.30	0.46	0.21	0.23	0.21
S-18	(TC)8	AAGAAGCCTTTCTTGCCTCC ATGAACCATCATGGGGAAAA	59.00	5	3.12	1.32	0.71	0.68	0.67
S-24	(TC)8	TGATGAAAATGGAGGGAAGC GGATCGAGTCCAAAATCGAA	58.00	3	1.13	0.27	0.10	0.11	0.20
S-43	(CT)9	TGATTGCGATAAATCCGACA TGTGAGGCTCGTGTTTCAAG	57.00	3	2.82	1.10	0.59	0.65	0.57
S-73	(ACA)6	AGCTGCCAGTTAGCCATGTT CCGGAAACTGAGTCATGGAT	58.00	3	1.58	0.59	0.46	0.37	0.31
S-77	(GCA)7	CAGAGCCATTGAAGCTCTCC ACACCAGACCTCTCACGACC	57.00	4	2.13	0.92	0.44	0.53	0.45
S-78	(GAA)6	AAACATGAACCTCCCATTCG CTGCCATGCTTGATACTGGA	57.00	3	1.99	0.81	0.26	0.50	0.60
S-84	(TCA)6	CTTTTCCAAGGGTCCAGTGA TCCTGAATCCCTATTCGTGC	57.00	3	1.61	0.70	0.31	0.38	0.35
S-92	(CAA)6	TGTAGGCATTTGTGGCAAGA TGTTTCAAATGCCAAGCAAA	57.00	3	1.96	0.85	0.55	0.49	0.49
S-94	(GAC)6	TTTGAGAGATGGGGGAACAC GAACAGGCTTTACAACCCCA	57.00	2	1.38	0.45	0.12	0.28	0.25
S-97	(ACC)6	GAATCGAAACTTTCCACCGA GCTCATTGCAACTACTGCCA	57.00	2	1.53	0.53	0.39	0.35	0.40
S-116	(TCA)7	ACCACATTCCCAAATTCCAA CTGTCAAATCGAGTGGCTCA	57.00	4	2.10	0.95	0.43	0.52	0.47
S-120	(GCA)6	AGGTGAACACGGTTCTTTGG TCCTCCCTATTTCTGGGCTT	57.00	4	1.26	0.40	0.13	0.21	0.19
S-132	(TCC)6	TCCTAAGCTCTGGTGGTGCT TGTGGGTCATAATGGTGGTG	57.00	3	1.59	0.72	0.29	0.37	0.34
S-135	(CTG)6	GGGAGAATCCTGAATCGACA CAACACTACCAAATGCCACG	56.00	3	1.26	0.43	0.17	0.21	0.19
S-155	(AG)8	CCCTCTTTGCTGTCATGGAT CAAAGGCAAACAAAAAGCGT	57.00	2	1.48	0.51	0.39	0.33	0.41
S-160	(AT)9	AAATTTGCCTATTCACCCCC AGACCGAGATTTGGTTCGTG	57.00	2	2.18	0.84	0.21	0.54	0.58
S-163	(CA)7	GCTGCAGTTTTACCAGAGCC AGGTGTGGGCATGTAGGAAG	57.00	4	1.54	0.72	0.25	0.35	0.36
S-165	(CA)8	AAGCTCACGATGGTGGTGAT ACGTCAAGCTGAGCAAGGTT	57.00	3	1.54	0.71	0.15	0.35	0.33
S-172	(CT)6	TCCTTGACTGGGAAATTCAAA TCAGCCAATCAATTCAATACCA	57.00	2	1.13	0.23	0.09	0.12	0.14
S-174	(CT)7	CCGACTTAAAACCCACTTCC CAAGCTATGCCAAGTGCGTA	57.00	2	2.00	0.69	0.19	0.50	0.57
S-179	(GA)10	GCAAAGCAACACATCAGCAT AGTTGAGGTATGGGGTGGTG	57.00	3	2.65	1.06	0.52	0.62	0.55
S-184	(AAG)6	ATGATGATGACGACGACGAA CGACAACAGCTCCAGAATCA	57.00	3	1.56	0.66	0.42	0.36	0.33
S-187	(ACA)5	GCCTCCCTTAAAACACTCCC CTAGCCTTTGCCCCTTCTCT	57.00	3	2.87	1.09	0.53	0.65	0.60
S-188	(ACA)5	ATGGAAACATGTGACCACCA AAACAGGGTCGATGGTTCTG	57.00	2	1.37	0.44	0.30	0.27	0.25
S-189	(ACC)5	TGCTGATGGCATGTAAGGAG CCGCACGAGGATAATTTTGT	57.00	2	1.28	0.46	0.07	0.22	0.21
S-210	(CAA)5	AGGGTTTGAAGGGTTGCTCT TGCAGTGAAAGCAACTGTGA	57.00	2	1.50	0.51	0.29	0.33	0.30
S-224	(AC)8	AAGCATCCATTGAAGAACCG CTCAGCACACACAGAGGGAA	57.00	2	1.15	0.25	0.10	0.13	0.22
S-239	(CGA)7	TCTGAAAGCACTGACCCTCC TGAAGCCATCATTCACAACC	57.00	4	2.65	1.13	0.53	0.62	0.56
S-286	(AG)7	CTTTCGTCTATGCAGCTCCC GGGTTGACCCACATCCCTAT	57.00	3	2.57	1.02	0.47	0.61	0.61
S-288	(TGT)5	GTTGCTCGCTTTTCGAAGTC AGCCAAGATGAAGAAAGGCA	57.00	4	2.50	1.13	0.43	0.60	0.57
Mean					1.83	0.71	0.33	0.40	0.40

Note: Tm, annealing temperature; *Na*, the number of alleles observed; *Ne*, the effective alleles; *Ho*, the observed heterozygosity; *He*, the expected heterozygosity; *I*, the Shannon’s information index; *PIC*, the polymorphic information content.

**Table 2 ijms-26-02346-t002:** Analysis of molecular variance (AMOVA) of 95 blackcurrant accessions based on population structure analysis results (K = 3).

Source	Degrees of Freedom	Sum of Square	Mean of Square	Est. Var.	%
Among clusters	2	50.728	25.364	0.293	5%
Between blackcurrant cultivars within clusters	92	629.193	6.839	0.877	14%
Between cultivars within blackcurrant cultivars	95	483.000	5.084	5.084	81%
Total	189	1162.921		6.254	100%

**Table 3 ijms-26-02346-t003:** Comparison of genetic diversity among core sets and initial collection.

Collection Type	*N*	*Na*	*A%*	*Ne*	*I*	*Ho*	*He*	*PIC*
Initial collection	95	2.935		1.749	0.652	0.335	0.374	0.354
PC	21	2.839	96.7%	1.756	0.664	0.335	0.381	0.361
CH	19	2.839	69.7%	1.770	0.656	0.336	0.378	0.361
IC	27	2.903	98.9%	1.771	0.665	0.336	0.381	0.370

Note: PC, core germplasm set extracted by Power Core; CH, core germplasm set extracted by Core Hunter; IC, integrated core collection; A%, allele retention rate.

**Table 4 ijms-26-02346-t004:** *Ribes* accessions collected from eight different origins.

NO.	Accessions Name	Species	Origin
1	Ben Nevis	*Ribes nigrum*	Britain
2	Baldwin	*Ribes nigrum*	Britain
3	Mendip Cross	*Ribes nigrum*	Britain
4	Big Ben	*Ribes nigrum*	Britain
5	Ben Gairn	*Ribes nigrum*	Britain
6	Ben Tirran	*Ribes nigrum*	Britain
7	Ben Lomond	*Ribes nigrum*	Britain
8	Ojebyn	*Ribes nigrum*	Sweden
9	Liangyehoupi	*Ribes nigrum*	Russia
10	Zusha	*Ribes nigrum*	Russia
11	Exotic	*Ribes nigrum*	Russia
12	Xielieqinaya	*Ribes nigrum*	Russia
13	Vologda	*Ribes nigrum*	Russia
14	Globus	*Ribes nigrum*	Russia
15	Gejinzige	*Ribes nigrum*	Russia
16	Adelinia	*Ribes nigrum*	Russia
17	Sophia	*Ribes nigrum*	Russia
18	Lama	*Ribes nigrum*	Russia
19	Belaruskaja	*Ribes nigrum*	Russia
20	Bagira	*Ribes nigrum*	Russia
21	Zwiezda	*Ribes nigrum*	Russia
22	Gezishiseng	*Ribes nigrum*	Russia
23	Kantata	*Ribes nigrum*	Russia
24	Nailor	*Ribes nigrum*	Russia
25	Primorskij pearl	*Ribes nigrum*	Russia
26	E-14	*Ribes nigrum*	Russia
27	E-15	*Ribes nigrum*	Russia
28	Baopifengchan	*Ribes nigrum*	Russia
29	Fertodi	*Ribes nigrum*	Poland
30	Orville	*Ribes nigrum*	Poland
31	Bagada	*Ribes nigrum*	Poland
32	Bona	*Ribes nigrum*	Poland
33	Roodknop	*Ribes nigrum*	Netherland
34	Risager	*Ribes nigrum*	Denmark
35	Black smith	*Ribes nigrum*	Canada
36	C17	*Ribes nigrum*	China
37	C19	*Ribes nigrum*	China
38	C28	*Ribes nigrum*	China
39	C11	*Ribes nigrum*	China
40	94-4-13	*Ribes nigrum*	China
41	17-29	*Ribes nigrum*	China
42	W1-2	*Ribes nigrum*	China
43	E16	*Ribes nigrum*	China
44	16A	*Ribes nigrum*	China
45	14(17-5)	*Ribes nigrum*	China
46	A16	*Ribes nigrum*	China
47	Hanfeng	*Ribes nigrum*	China
48	19C	*Ribes nigrum*	China
49	18C	*Ribes nigrum*	China
50	17B	*Ribes nigrum*	China
51	13C	*Ribes nigrum*	China
52	14C	*Ribes nigrum*	China
53	15C	*Ribes nigrum*	China
54	16C	*Ribes nigrum*	China
55	17C	*Ribes nigrum*	China
56	Aw-2	*Ribes nigrum*	China
57	15-3	*Ribes nigrum*	China
58	17-29-1	*Ribes nigrum*	China
59	15-4	*Ribes nigrum*	China
60	BW-2	*Ribes nigrum*	China
61	15-2	*Ribes nigrum*	China
62	15-1	*Ribes nigrum*	China
63	Aw-4	*Ribes nigrum*	China
64	Aw-3	*Ribes nigrum*	China
65	Yade-3	*Ribes nigrum*	China
66	17-29-3	*Ribes nigrum*	China
67	15-8	*Ribes nigrum*	China
68	Aw-1	*Ribes nigrum*	China
69	15-10	*Ribes nigrum*	China
70	17-29-2	*Ribes nigrum*	China
71	17-29-5	*Ribes nigrum*	China
72	15-6	*Ribes nigrum*	China
73	BW-3	*Ribes nigrum*	China
74	Yade-1	*Ribes nigrum*	China
75	Yade-2	*Ribes nigrum*	China
76	15-5	*Ribes nigrum*	China
77	Bw-1	*Ribes nigrum*	China
78	Lw-1	*Ribes nigrum*	China
79	15-9	*Ribes nigrum*	China
80	17-29-4	*Ribes nigrum*	China
81	Yade-4	*Ribes nigrum*	China
82	0A14	*Ribes nigrum*	China
83	Bw-4	*Ribes nigrum*	China
84	15-7	*Ribes nigrum*	China
85	SU-3	*Ribes nigrum*	China
86	Suiyanyihao	*Ribes nigrum*	China
87	Suiyanerhao	*Ribes nigrum*	China
88	Danjianghei	*Ribes nigrum*	China
89	Suanpanzi	*Ribes nigrum*	China
90	Muxuan 2008-6	*Ribes nigrum*	China
91	Muxuan 2012-6	*Ribes nigrum*	China
92	Muxuan 2015-10	*Ribes nigrum*	China
93	Muxuan 2011-14	*Ribes nigrum*	China
94	Muxuan 2013-10	*Ribes nigrum*	China
95	Muxuan 2015-13	*Ribes nigrum*	China
96	Crusader	*Ribes rubrum*	Russia
97	ER-1	*Ribes rubrum*	Russia
98	Maer	*Ribes rubrum*	Russia
99	Red Spring	*Ribes rubrum*	Russia
100	E-RED	*Ribes rubrum*	Russia
101	Cherry	*Ribes rubrum*	Poland
102	Witte Parel	*Ribes rubrum*	French
103	Ussuri	*Ribes ussuriensis*	Russia
104	Pixwell	*Ribes uva-crispa*	Russia
105	Hongyeheidou	*Ribes americanum*	America
106	Xiangchabiaozi	*Ribes odoratum*	China
107	Xinganchabiao	*Ribes panciflorum*	China

## Data Availability

Original contributions to this research are included in the article; please contact the corresponding authors for further inquiries.

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
