# Peer review of "Genetic Diversity Assessment and Core Germplasm Screening of Blackcurrant (Ribes nigrum) in China via Expressed Sequence Tag–Simple Sequence Repeat Markers"

_ijms, 2025, doi:10.3390/ijms26052346_

Round 1
Reviewer 1 Report
Comments and Suggestions for Authors
In This study, 31 EST-SSR markers were used to evaluate the genetic diversity and phylogenetic relationships of 95 blackcurrant cultivars and 12 other Ribes accessions, and the core germplasm of blackcurrant was screened. All the data analysis is standard, and this finding is interesting. However, I feel that some experimental results need to be explained in detail or some texts should be revised. The details are as follows:
1. Please note the difference between ‘cultivar’ and ‘accessions’. In L16, I suggest changing '12 other Ribes cultivars' to '12 other Ribes accessions’. Similar problems exist in other texts.
2. In L25-26, there is no specific contribution of this study to the genetics and breeding of blackcurrant.
3. In L90, I suggest a brief description of the transcriptome data here.
4. In L124-146, whether pedigree relationships can be seen in the breeding material, these contents need to be evaluated. For example, the diversification or uniformity of the pedigree of Chinese bred varieties.
5. In L161-162, what reason or special feature of 'baopifengcha' has that makes it the most genetically distant from other breeds, and it needs to be added.
6. In L177, the resolution of Figure 3 is too low and the sample name is not clear.
7. In L181, 'the genetic diversity of the NEAU collection ', 'NEAU'?
8. In L182, need to describe what data the core germplasm screening is based on.
9. In L201, figure 4 is not self-explanatory, PC, CH, IC representation is unclear.
10. In discusion part, the discussion is too simplistic, and a more detailed comparative analysis should be carried out. For example: To compared other countries, how the genetic diversity levels of existing blackcurrant cultivars in China? What are the specific phylogenetic relationships between these blackcurrant cultivars and other Ribes accessions? How is this different of this results from existing research? What new insights were gained in this study?
11. In L214-215, 'However, the SSR markers often have a high rate of redundancy when used for analysis', how this relates to the features of this study should be further clarified.
12. In L229-230, 'This difference may be related to variations in the markers used and the samples analyzed' should be described in more detail. Are the markers of this study not representative, or are the source of the accessions too single and poor diversity?
13. In L236, what are 'major historical events'? It should be described in detail.
14. In L237, this core germplasm set needs to be described and compared in detail. The loss of diversity is identified in terms of visual evaluation of accessions or forms, such as the screening of certain repetitive germplasm mentioned in the introduction.
15. In L251, whether this study can combine the known original information, breeding records, and genetic participation of local wild species to speculate the accurate source or parental relationship of some accessions, and analyze the genetic diversity and the formation of population structure of existing Chinese blackcurrant germplasm. It needs to be discussed in detail.
Author Response
Response to Reviewer Comments
We greatly appreciate the thorough and thoughtful comments provided on our submitted article. We made sure that each one of the reviewer comments has been addressed carefully and the paper is revised accordingly.
Comments 1: Please note the difference between ‘cultivar’ and ‘accessions’. In L16, I suggest changing ‘12 other Ribes cultivars’ to ‘12 other Ribes accessions’. Similar problems exist in other texts.
Response 1: Sorry for our misunderstanding and thank you for your review. We have revised the other ‘Ribes cultivars’ to ‘Ribes accessions’ based on your suggestion, except for blackcurrant. Please see L15, 16, 55, 89, 113, 138, 140, 180, 187, 259, 307, 360, 365, and 366 of the file: "ijms-3404105-Manuscript_ trackchanges.docx" for details.
Comments 2: In L25-26, there is no specific contribution of this study to the genetics and breeding of blackcurrant.
Response 2: The core germplasm resources established in this study primarily provide a reference standard for genetic breeding of blackcurrant. Additionally, the primers we developed can offer more primer options for Ribes accessions. To clarify our intent, we have added ‘especially Ribes nigrum’ after Ribes. Please see L25-27 of the file: "ijms-3404105-Manuscript_ trackchanges.docx" for details.
Comments 3: In L90, I suggest a brief description of the transcriptome data here.
Response 3: We have followed your suggestion and provided a brief description of the transcriptomic data obtained from the NCBI database, prior to presenting the results of SSR loci identification. Please see L93-94 of the file: "ijms-3404105-Manuscript_ trackchanges.docx" for details.
Comments 4: In L124-146, whether pedigree relationships can be seen in the breeding material, these contents need to be evaluated. For example, the diversification or uniformity of the pedigree of Chinese bred varieties.
Response 4: We fully recognize the importance of pedigree relationships in breeding materials. Unfortunately, due to environmental factors, some of the early-introduced cultivars have perished, and only the parent plants of two cultivars have been preserved in their entirety. The limited data available is insufficient for meaningful reference, making it difficult to present them effectively.
Comments 5: In L161-162, what reason or special feature of 'baopifengcha' has that makes it the most genetically distant from other breeds, and it needs to be added.
Response 5: We have included a description of the unique characteristics of the ‘Baopifengchan’ cultivar in the discussion. Please see L253-256, of the file: "ijms-3404105-Manuscript_ trackchanges.docx" for details.
Comments 6: In L177, the resolution of Figure 3 is too low and the sample name is not clear.
Response 6: We have adjusted the resolution of Figure 3. Following the suggestions of other reviewers, we have swapped the contents of sections 2.3 and 2.4 in the Results. The image is now labeled as Figure 2. Please see L154 of the file: "ijms-3404105-Manuscript_ trackchanges.docx" for details.
Comments 7: In L181, ‘the genetic diversity of the NEAU collection’, ‘NEAU’?
Response 7: For the sake of readability, we have moved the description of ‘the NEAU collection’ from the Materials and Methods section to the Introduction. Please see L54-56 of the file: "ijms-3404105-Manuscript_ trackchanges.docx" for details.
Comments 8: In L182, need to describe what data the core germplasm screening is based on.
Response 8: We have added a description related to the core germplasm selection based on genetic markers, matrices, and the genetic parameters of population data in L199-200 of the file: "ijms-3404105-Manuscript_ trackchanges.docx".
Comments 9: In L201, Figure 4 is not self-explanatory, PC, CH, IC representation is unclear.
Response 9: In response to your suggestion, we have added a figure legend for Figure 4. Please see L218-222 of the file: "ijms-3404105-Manuscript_ trackchanges.docx" for details.
Comments 10: In discusion part, the discussion is too simplistic, and a more detailed comparative analysis should be carried out. For example: To compared other countries, how the genetic diversity levels of existing blackcurrant cultivars in China? What are the specific phylogenetic relationships between these blackcurrant cultivars and other Ribes accessions? How is this different of this results from existing research? What new insights were gained in this study?
Response 10: In the Results section, we have provided a further description of the phylogenetic relationships between blackcurrant cultivars and other 12 Ribes accessions, as well as the parent structure of the 107 Ribes accessions. Additionally, in the Discussion, we have revisited this result and compared it with the interspecific phylogenetic relationships reported in the paper. Please see L181-189, 261-263, and 288-291 of the file: "ijms-3404105-Manuscript_ trackchanges.docx" for details.
Comments 11: In L214-215, 'However, the SSR markers often have a high rate of redundancy when used for analysis', how this relates to the features of this study should be further clarified.
Response 11: We have rephrased and clarified this statement. Please see L232-234 of the file: "ijms-3404105-Manuscript_ trackchanges.docx" for details.
Comments 12: In L229-230, 'This difference may be related to variations in the markers used and the samples analyzed' should be described in more detail. Are the markers of this study not representative, or are the source of the accessions too single and poor diversity?
Response 12: Please see We have revisited the discussion regarding the mean values of He and Ho for the newly developed EST-SSR markers being lower than those for the SSR markers. The updated results can be found in L236-244 of the file: "ijms-3404105-Manuscript_trackchanges.docx".
Comments 13: In L236, what are 'major historical events'? It should be described in detail.
Response 13: Through a review of the literature, we found that significant historical events, such as extreme cold weather and large-scale volcanic eruptions, can indeed impact population genetic diversity. However, these factors are not relevant to the focus of this study. Therefore, we have removed the related content.
Comments 14: In L237, this core germplasm set needs to be described and compared in detail. The loss of diversity is identified in terms of visual evaluation of accessions or forms, such as the screening of certain repetitive germplasm mentioned in the introduction.
Response 14: With reference to the RICESCO project, we have described the limitations of the derived core germplasm set and outlined the subsequent work aimed at enriching the core germplasm resources by incorporating phenotypic traits. The updated results can be found in L269-275, and 278-281 of the file: "ijms-3404105-Manuscript_trackchanges.docx".
Comments 15: In L251, whether this study can combine the known original information, breeding records, and genetic participation of local wild species to speculate the accurate source or parental relationship of some accessions, and analyze the genetic diversity and the formation of population structure of existing Chinese blackcurrant germplasm. It needs to be discussed in detail.
Response 15: We were highly interested in exploring the formation of the population structure of Chinese blackcurrants (Ribes nigrum) using known historical records, breeding documentation, and genetic contributions from local wild species. Unfortunately, we were only able to locate five incomplete breeding records, which are insufficient to support such an analysis at this time. In the future, we plan to investigate the population formation of Chinese blackcurrants by integrating phenotypic data with sequencing approaches.
We believe that the revisions made in response to your comments have significantly improved the manuscript. We are grateful for your constructive feedback and hope that the revised version meets your expectations.

Reviewer 2 Report
Comments and Suggestions for Authors
1. Overall Evaluation
This paper proposes a solution to solve the problem of lack of means of identification of black currant cultivars by developing new molecular markers. The experiment has a good experimental design and reliable data analysis. The study contributes to the identification of blackcurrant cultivars. However, there are some minor issues regarding the clarity of some parts and the presentation of the results. I recommend acceptance of the paper with minor revisions.
2. Strengths
As an important cash crop, existing research on blackcurrant is mainly based on nutritional identification and lacks efficient and reproducible means of varietal identification. This article is highly relevant to current research in the field and provides a marker basis for advancing the molecular identification of black currant, which may stimulate further research.
3. Minor Issues and Suggestions for Improvement
Figure 1b has a dense distribution of horizontal coordinates, which may make it difficult for the reader to read, and I suggest increasing the distance between the horizontal axis labels. Suggested changes.
Line 58 There is an error in the numbering of the references, please correct it.
Line 121, 162 Ensure correct punctuation usage and font consistency. Suggested changes.
Line 182 Please indicate the full name of NEAU.
Line 209 The paragraph focuses on the interspecific relationships of the genus Ribes, and for ease of reading, it is recommended that the Latin name of the “Sophia” cultivar be indicated.
Line 300-302 Recommendation to harmonize the presentation of time units.
It is suggested that Table 4 be changed to a supplementary table because of its length.
After the suggested revisions, I believe this paper would be suitable for publication in its current form.
Author Response
Response to Reviewer Comments
We greatly appreciate the thorough and thoughtful comments provided on our submitted article. We apologize to the reviewer for our misunderstanding. We made sure that each one of the reviewer comments has been addressed carefully and the paper is revised accordingly.
Comments 1: Figure 1b has a dense distribution of horizontal coordinates, which may make it difficult for the reader to read, and I suggest increasing the distance between the horizontal axis labels. Suggested changes.
Response 1: Following your suggestion, we have revised the x-axis of Figures 1a &1b to improve the clarity and readability of the images. The updated results can be found in L107 of the file: "ijms-3404105-Manuscript_trackchanges.docx".
Comments 2: Line 58 There is an error in the numbering of the references, please correct it.
Response 2:We sincerely apologize for the oversight in our writing. The issues have been addressed and corrected accordingly. The updated results can be found in L59 of the file: "ijms-3404105-Manuscript_trackchanges.docx".
Comments 3: Line 121, 162 Ensure correct punctuation usage and font consistency. Suggested changes.
Response 3:We appreciate your identification of the errors in the manuscript, and we have now corrected them accordingly. The updated results can be found in L137, 168 of the file: "ijms-3404105-Manuscript_trackchanges.docx".
Comments 4: Line 182 Please indicate the full name of NEAU
Response 4:We have moved the description of ‘the NEAU collection’ from the Materials and Methods section to the Introduction. The updated results can be found in L154, of the file: "ijms-3404105-Manuscript_ trackchanges.docx".
Comments 5: Line 209 The paragraph focuses on the interspecific relationships of the genus Ribes, and for ease of reading, it is recommended that the Latin name of the “Sophia” cultivar be indicated.
Response 5:In accordance with your suggestion, we have supplemented the Latin name for ‘Sophia’ in the manuscript. The updated results can be found in L151 of the file: "ijms-3404105-Manuscript_ trackchanges.docx".
Comments 6: It is suggested that Table 4 be changed to a supplementary table because of its length.
Response 6:Table 4 in the manuscript primarily provides information on the names, species, and origins of 107 Ribes accessions. This information is crucial for the descriptions of the corresponding varieties in Sections 2.3 and 2.4 of the Results. Therefore, we have retained this table in the main text for the time being.
We sincerely appreciate the time and effort you have dedicated to reviewing our manuscript. Your insightful comments have been invaluable in enhancing the quality of our work. We have carefully addressed each of your suggestions and hope that the revisions meet your expectations. Thank you once again for your constructive feedback, which has significantly strengthened our paper.

Reviewer 3 Report
Comments and Suggestions for Authors
Dear authors, I have read your manuscript titled Genetic diversity assessment and core-germplasm screening of blackcurrant (Ribes nigrum) in China via EST-SSR markers. The manuscript is scientifically sound and for the most part well-written.
I do have some comments and/or questions for you, that need to be addressed before it is suitable for publishing. The minor issues and comments are in the PDF file attached to this review, and the following list contains the major concerns:
- You need to be consistent in your use of terminology to avoid confusion for the reader. The most pressing terms are “population”, “clade” and “specimen”. Each of them is a specific biological term that tends to be misused. I don’t think “population” is appropriate for your case study. I would suggest using the term “group” (e.g. group of cultivars or accessions). The term “clade” is phylogenetical term and means something very distinctly. I think you should use the term “genetic cluster” or “gene pool” (when describing the results of Structure). In Table 4 you use the term “specimen”. If you’re referring to samples, please use the term “sample” instead, but I believe the column in question refers to cultivars, am I right?
- In the last section of the Introduction you state your aims. However, the first aim should be: developing EST-SSR markers using the published transcriptome data for blackcurrant
- Since the Results section is before the Materials and methods section, a little more context should be given about methodology for your results. Otherwise, your results are unclear.
- Figure 1C – although the design of the 3D graph is good, due to the tick marks on the x-axis being adjacent to the bars makes the graph more difficult to read. If you can’t correct the placement of the tick marks, I would suggest using the standard stacked bar for this part of the figure.
- What is your reasoning for conductign some analyses on only the blackcurrant samples, and some on all samples? I would advise to be consistent, run the structure analysis on all samples, as well as AMOVA (using blackcurrant as one group, R. rubrum another, and the rest as a third group based on NJ analysis). This would enrich your research, and test the results obtained by only one analysis.
- I would suggest switching the order of sections 2.3 and 2.4 (genetic structure and AMOVA are more advanced analyses that are usually reported last in this type of papers)
- Please correct the Table 2. This seems to be a default output of the software, and categories should be renamed properly.
- Discussion needs to be improved significantly. While the 3.3 section is well written, sections 3.1 and 3.2 read more like overview of other research with no comparison to your research. Since you cite multiple similar papers, some comparisons could be made.
- In the Materials and methods section some improvements could be made by removing sentences that are more appropriate for the Introduction section, and adding some summary information on the geographical origin of accessions in the text. In the statistical subsections, please add information about the AMOVA (software and parameters) as well as additional information about PCoA (based on what data was performed).
- In this research you have essentially developed markers not only for blackcurrant, but also for several other species in the genus Ribes, method that is known as cross-amplification, but you don’t emphasize this enough. Especially, because cross-amplification in closely related species is often unsuccessful. This is apparent the most in the Conclusion section where you mention it in the final sentence. I would advise to expand on this in the paper more, to broaden the scope of your paper to researchers focusing on other Ribes species.

I have highlighted some grammatical or language issues in the PDF that you should correct and avoid using further. Please review and correct where you think appropriate.
Author Response
Response to Reviewer Comments
We greatly appreciate the thorough and thoughtful comments provided on our submitted article. We made sure that each one of the reviewer comments has been addressed carefully and the paper is revised accordingly.
Comments 1: You need to be consistent in your use of terminology to avoid confusion for the reader. The most pressing terms are “population”, “clade” and “specimen”. Each of them is a specific biological term that tends to be misused. I don’t think “population” is appropriate for your case study. I would suggest using the term “group” (e.g. group of cultivars or accessions). The term “clade” is phylogenetical term and means something very distinctly. I think you should use the term “genetic cluster” or “gene pool” (when describing the results of Structure). In Table 4 you use the term “specimen”. If you’re referring to samples, please use the term “sample” instead, but I believe the column in question refers to cultivars, am I right?
Response 1:Thank you very much for pointing out the descriptive errors regarding ‘population’, ‘clade’, and ‘specimen’ in the manuscript. Based on your suggestions, we have made corrections to the relevant sections. Since there was no issue with the division of populations in the text, we ultimately decided to describe the groups as ‘Cluster’ and ‘Group’. The updated results can be found in L163-189 of the file: "ijms-3404105-Manuscript_ trackchanges.docx".
Comments 2: In the last section of the Introduction you state your aims. However, the first aim should be: developing EST-SSR markers using the published transcriptome data for blackcurrant.
Response 2:Based on your suggestion, we have added: ‘EST-SSR: develop new SSR markers that are universally applicable to crops within the Ribes genus’ as the first aim in the last section of the Introduction. The updated results can be found in L86-87 of the file: "ijms-3404105-Manuscript_ trackchanges.docx".
Comments 3: Since the Results section is before the Materials and methods section, a little more context should be given about methodology for your results. Otherwise, your results are unclear.
Response 3:To enhance the readability of the manuscript, we have followed your suggestion by presenting the source of the transcriptome data in Section 2.1 of the Results. Additionally, the description of the NEAU collection has been relocated to the Introduction section. The updated results can be found in L54-56 and 93-94 of the file: "ijms-3404105-Manuscript_ trackchanges.docx".
Comments 4: Figure 1C – although the design of the 3D graph is good, due to the tick marks on the x-axis being adjacent to the bars makes the graph more difficult to read. If you can’t correct the placement of the tick marks, I would suggest using the standard stacked bar for this part of the figure.
Response 4:The 3D graph in Figure 1C provides a more intuitive representation of EST-SSR locus information and the number of repeat types for different motifs compared to a standard stacked bar chart. However, the results remained unclear. Therefore, we have converted Figure 1C into Supplementary Table 2. The updated results can be found in Table S2.
Comments 5: What is your reasoning for conductign some analyses on only the blackcurrant samples, and some on all samples? I would advise to be consistent, run the structure analysis on all samples, as well as AMOVA (using blackcurrant as one group, R. rubrum another, and the rest as a third group based on NJ analysis). This would enrich your research, and test the results obtained by only one analysis.
Response 5:Based on your suggestion, in the Results section, we have provided a further description of the phylogenetic relationships between blackcurrant cultivars and 12 other Ribes accessions, as well as the parent structure of the 107 Ribes accessions. Additionally, in the Discussion, we have revisited this result and compared it with the interspecific phylogenetic relationships reported in the paper. However, due to the large difference in the number of accessions within the two groups, which reduces the reliability of the AMOVA analysis results, we have not added this to the manuscript. Please see L181-189, 261-263, 288-291, Table S5-6, and Figure S1 of the file: 'ijms-3404105-Manuscript_ trackchanges.docx' for details.
Comments 6: I would suggest switching the order of sections 2.3 and 2.4 (genetic structure and AMOVA are more advanced analyses that are usually reported last in this type of papers)
Response 6:In accordance with your suggestion, we have swapped the content of Sections 2.3 and 2.4 in the Results. Please see L128-189 of the file: 'ijms-3404105-Manuscript_ trackchanges.docx' for details.
Comments 7: Please correct the Table 2. This seems to be a default output of the software, and categories should be renamed properly.
Response 7:Fllowing your suggestion, we have revised "Pops" to "Clusters" in Table 2. Please see L194-196 of the file: 'ijms-3404105-Manuscript_ trackchanges.docx' for details.
Comments 8: Discussion needs to be improved significantly. While the 3.3 section is well written, sections 3.1 and 3.2 read more like overview of other research with no comparison to your research. Since you cite multiple similar papers, some comparisons could be made.
Response 8:In accordance with your suggestions, we have further discussed the characteristics of the EST-SSR markers, the genetic relationships among the 107 Ribes accessions, and the limitations of the core germplasm resource construction in the manuscript. Please see L232-234, 236-244, 269-275, 278-281, 288-291 of the file: 'ijms-3404105-Manuscript_ trackchanges.docx' for details.
Comments 9: In the Materials and methods section some improvements could be made by removing sentences that are more appropriate for the Introduction section, and adding some summary information on the geographical origin of accessions in the text. In the statistical subsections, please add information about the AMOVA (software and parameters) as well as additional information about PCoA (based on what data was performed).
Response 9:Following your suggestions, we have revised the Materials and Methods section. Specifically, we have relocated the description of the Ribes accessions collection to the Introduction section and supplemented the details regarding the software used for AMOVA analysis () and the data types employed for constructing the PCoA mapping. Please see L232-234, 236-244, 269-275, 278-281, and 288-291 of the file: 'ijms-3404105-Manuscript_ trackchanges.docx' for details.
Comments 10: In this research you have essentially developed markers not only for blackcurrant, but also for several other species in the genus Ribes, method that is known as cross-amplification, but you don’t emphasize this enough. Especially, because cross-amplification in closely related species is often unsuccessful. This is apparent the most in the Conclusion section where you mention it in the final sentence. I would advise to expand on this in the paper more, to broaden the scope of your paper to researchers focusing on other Ribes species.
Response 10:Regarding the issue you raised about cross-amplification of the newly developed primers in related species, we have discussed and summarized the population structure of blackcurrant and the population structure and genetic relationships within the Ribes genus in the Discussion and Conclusion sections. Please see L253-256, 261-263, 356-357, 357-359, and 362-363 of the file: 'ijms-3404105-Manuscript_ trackchanges.docx' for details.
Thank you for your meticulous review and the thoughtful comments you provided on our manuscript. Your feedback has been crucial in refining our research and ensuring its alignment with the highest scholarly standards. We have thoroughly revised the manuscript in response to your suggestions and hope that the changes reflect our commitment to excellence. Your support and guidance are greatly appreciated.

Reviewer 4 Report
Comments and Suggestions for Authors
Dear Authors,
see the pdf document below.

Author Response
Response to Reviewer Comments
We greatly appreciate the thorough and thoughtful comments provided on our submitted article. We made sure that each one of the reviewer comments has been addressed carefully and the paper is revised accordingly.
Comments 1: L 110, Probably, this is wrong, and it stands for Table 1, not Table 2.
Response 1: Following your reminder, we have made the necessary corrections. Please see L113 of the file: 'ijms-3404105-Manuscript_ trackchanges.docx' for details.
Comments 2: L118, There is no explanation about Table 1 in the text. What no stands for?
Response 2: Following your reminder, we have revised "No." to "Primer". Please see L125 of the file: 'ijms-3404105-Manuscript_ trackchanges.docx' for details.
Comments 3: L211, delete I in front of The
Response 3: We have made the necessary corrections according your reminder. Please see L229 of the file: 'ijms-3404105-Manuscript_ trackchanges.docx' for details.
Comments 4: White currant missing hybrid parentage from Ribes rubrum (red currant)
Response 4: According to the internationally recognized systematic classification of the Ribes genus, the Latin names for white currants and red currants have not yet been distinguished. Therefore, their Latin names remain the same, and we have not made any modifications to them at this time.
Comments 5: L306, please correct ‘GenAlex’ to ‘GenAlEx’.
Response 5: We have corrected ‘GenAlex’ to ‘GenAlEx’ in L306, 343, 350 of the file: 'ijms-3404105-Manuscript_ trackchanges.docx' for details.
Coomments 6: M&M are scientifically valid written, clear and concise; and the laterrmost softwares/tools were used in population genetics, evolution, and phylogenetics analysis in this paper.
Response 6: The PCoA mapping was generated using an online tool, and the specific calculation methods were not provided. Therefore, we have revised the sentence based on the uploaded data. Please see L351-352 of the file: 'ijms-3404105-Manuscript_ trackchanges.docx' for details.
We sincerely apologize for the oversight and any inconvenience caused by the errors in our initial submission. We deeply appreciate your patience and the opportunity to correct these mistakes. Your feedback has been invaluable in helping us improve the quality and accuracy of our work. We have carefully addressed all the issues raised and hope that the revised manuscript now meets the expected standards. Thank you for your understanding and for guiding us toward a more polished and rigorous presentation of our research.

Round 2
Reviewer 3 Report
Comments and Suggestions for Authors
Thank you for considering the comments from the first round of the review. There are still a couple of minor issues that need to be resolved:
- The term “population” is not appropriate to use in this research. You did not have natural populations in your sample, nor artificial ones, rather your samples were accessions. Same goes for “population structure analysis”. Although Structure is used dominantly in population analyses, here you effectively used the method to analyse your accessions. When presenting the results and referring to major groups that Structure produced, please refer to them as gene pools or ancestral populations to make a distinction from the term generic population.
- From my understanding of the text, as inputs for AMOVA you have grouped your accessions into groups based on the gene pools in Structure analysis. First be consistent in usage of groups/clusters in the text (in presenting the results of Structure for 95 blackcurrant accessions you use clusters, but when presenting results of all accessions you use groups). Chose one term and stick with it. Secondly, in the table, the categories still need to be improved. “Among clusters” is fine. But “Among individuals” refers to the smaller groups within each cluster (in your case different Ribes species) so I suggest “Between Ribes species within clusters”. Instead of “Within individuals” should stand “Between Accessions within Ribes species”.
- You have not addressed the usage of term “Specimen” in the Table 4. Are those the names of actual cultivars? Please correct if so.
Author Response
Response to Reviewer Comments
We are grateful for your continued attention to our manuscript and for raising additional questions. Your comments have provided us with an opportunity to clarify and refine our study. Please find our detailed responses below, along with the corresponding revisions in the manuscript.
Comments 1: The term “population” is not appropriate to use in this research. You did not have natural populations in your sample, nor artificial ones, rather your samples were accessions. Same goes for “population structure analysis”. Although Structure is used dominantly in population analyses, here you effectively used the method to analyse your accessions. When presenting the results and referring to major groups that Structure produced, please refer to them as gene pools or ancestral populations to make a distinction from the term generic population.
Response 1: Since the experimental materials consist of collected and bred blackcurrant cultivars, which do not belong to a natural population, we have revised the description of “the blackcurrant population” to “the blackcurrant gene pool” in accordance with your suggestion. Please refer to L20, 24, 148, 162, 171, 219, and 258 in ijms-3404105-Manuscript_ trackchanges_2.docx for details. Additionally, we have revised “Original population” to “initial collection” in L212, Table 3, and Figure 4.
Comments 2: From my understanding of the text, as inputs for AMOVA you have grouped your accessions into groups based on the gene pools in Structure analysis. First be consistent in usage of groups/clusters in the text (in presenting the results of Structure for 95 blackcurrant accessions you use clusters, but when presenting results of all accessions you use groups). Chose one term and stick with it. Secondly, in the table, the categories still need to be improved. “Among clusters” is fine. But “Among individuals” refers to the smaller groups within each cluster (in your case different Ribes species) so I suggest “Between Ribes species within clusters”. Instead of “Within individuals” should stand “Between Accessions within Ribes species”.
Response 2: Thank you for your suggestions. We have unified “Group G1-2” and “Group a-g” into “Cluster â… -â…¨” (L182-186, Figure S1, Table S5-6). Additionally, we have revised the classifications in the AMOVA analysis. Following your recommendation, we have changed “Among individuals” to “Between blackcurrant cultivars within clusters” and “Within individuals” to “Between cultivars within blackcurrant cultivars”. These changes have been incorporated into the main text (Please refer to L162, 171, and 258 in ijms-3404105-Manuscript_ trackchanges_2.docx for details).
Comments 3: You have not addressed the usage of term “Specimen” in the Table 4. Are those the names of actual cultivars? Please correct if so.
Response 3: We apologize for our oversight. We have corrected “Specimen” to “Accession Name” in Table 4 (Please refer to L309 in ijms-3404105-Manuscript_ trackchanges_2.docx for details).
Thank you once again for your thorough review and valuable input. We believe that your comments have greatly improved the quality of our manuscript. We hope that our responses and revisions meet your expectations, and we are happy to provide any additional information if needed.
